# SSPictR: Spatial Semantic Pointer Picture Representation

## Abstract

The development of image representations that capture semantic and spatial information efficiently, which are also interpretable and generalisable, remains unsolved. Drawing from a cognitive modelling framework, we propose *SSPictR* – a biologically plausible image representation based on spatial semantic pointers (SSPs). SSPictR encodes semantic labels and their spatial locations extracted from segmentation maps and only requires a single vector to capture a fully decodable neuro-symbolic representation of a natural scene. It is inherently interpretable, offers a high compression factor and significantly faster inference speed on downstream tasks, such as scene recognition. We evaluate the efficiency and generalisability of SSPictR on the popular Places365, and ADE20K datasets for scene recognition, on COCOStuff for segmentation reconstruction, and on VISC and Savoias for prediction of visual complexity. We show that the scene representations provided by SSPictR are more generalisable within and across these tasks while only requiring a fraction of model parameters and, therefore, offer 25 times higher inference speed, with comparable accuracy. As such, SSPictR opens up a new direction for future research on cognitively-inspired image representations that are not only significantly smaller but also more interpretable and generalisable.

## 1 Introduction

Machine learning models for computer vision have achieved human-level performance on a number of tasks, such as object detection (Zong et al., 2023), semantic segmentation (Chen et al., 2022), or depth estimation (Zhao et al., 2023). But these improvements come at an ever-increasing cost: These models are highly inefficient and lack interpretability, i.e. they require training of millions of parameters from large amounts of data and their learned internal representations are notoriously challenging to analyse and understand. Furthermore, despite impressive performance on tasks that they were trained for, they still lack generalisability to out-of-domain (OOD) data (Geirhos et al., 2018; Mahner et al., 2024). Aligning model and human representations has increased robustness in object detection (Geirhos et al., 2019), improved performance (Sucholutsky et al., 2023), and has advanced our understanding of human cognition through better computational models (Mahner et al., 2024). While there has been a lot of research in aligning deep neural networks (DNNs) in object classification tasks (Mahner et al., 2024; Muttenthaler et al., 2022; Geirhos et al., 2021), human alignment for scene understanding remains under-explored (Bartnik & Groen, 2024).

Computational models of human perception have a long-standing history of research in cognitive science (Kotseruba & Tsotsos, 2020). Most interesting are biologically-plausible cognitive models, as they can be naturally integrated with DNNs while maintaining the interpretability of symbolic systems. A popular model is the semantic pointer architecture (SPA; Eliasmith 2013) – a cognitive modelling framework based on vector symbolic algebras (VSAs). VSAs use hyperdimensional distributed vectors for efficient and robust representation of concepts (Kleyko et al., 2022). There are many successful applications of cognitively-inspired VSAs, for example, in abstract reasoning (Hersche et al., 2023), ego-motion prediction (Mitrokhin et al., 2019), path integration (Dumont et al., 2022), and reinforcement learning (Bartlett et al., 2022). Particularly interesting as representations for scene understanding are the works by Komer et al. (2019) and Penzkofer et al. (2024), as they encode objects in a grid-like continuous vector space similar to cognitive maps found in the brain (Bermudez-Contreras et al., 2020).

Figure 1: Overview of SSPictR – a cognitively-inspired image representation that only requires a single vector, which can be used to predict scene categories, visual complexity ratings, or even decode a full cognitive map of the scene.

In this work, we bring together this research and propose SSPictR– a neurally inspired image representation that is more efficient, interpretable, and generalisable than existing image representations. At its core, SSPictR encodes objects from segmentation maps into a continuous vector space that compresses the semantic and spatial information of a scene. Figure 1 shows an overview. In contrast to existing representations, SSPictR offers a significantly better compression factor of only 0.46 bits per pixel (vs 8 bits for segmentation maps) and an inference speed of 2,857 fps for scene recognition. The compressed vector representation only has 3,751 dimensions and can be queried, used to fully reconstruct the segmentation map, or used as a feature embedding for further downstream tasks, such as visual complexity prediction. We evaluate different encoding methods for this novel representation based on reconstruction error of the segmentation map. We also show that SSPictR is highly generalisable across tasks (scene recognition, visual complexity) and datasets, providing competitive performance for OOD scene recognition on the ADE20K dataset (Zhou et al., 2017).

In summary, we present a novel scene representation: *SSPictR*. SSPictR is highly efficient, interpretable, and generalisable. We provide a construction method from segmentation maps and evaluate different encoding schemes. Further, we evaluate the representation quality via linear probing on scene recognition, its generalisability on OOD data, and highlight potential applications in computer vision, cognitive science, and robotics.

## 2 RELATED WORK

### 2.1 SCENE REPRESENTATIONS

The choice of data representation is a key factor for the performance of machine learning models (Bengio et al., 2013). Scene representation, in particular, is challenging due to complex configurations, i.e., scenes are comprised of diverse objects in complex spatial layouts with substantial semantic ambiguity (Xie et al., 2020). Furthermore, scenes can also be characterised as environments for embodied agents to navigate in (Malcolm et al., 2016).

**Scene Recognition.** Scene recognition, the task of classifying scenes into categories, is considered a fundamentally important but challenging task in computer vision (Zeng et al., 2021), with a wide range of applications, from robot navigation (Yadav et al., 2023) to disaster detection (Muhammad et al., 2018). Current benchmark datasets include Places365 (Zhou et al., 2018), ADE20K (Zhou et al., 2017), SUN397 (Xiao et al., 2010), and MIT67 (Quattoni & Torralba, 2009). While MIT67 only contains 15 thousand images classified into 67 indoor scenes, the most recent Places365, consists of 10 million images with annotations for 434 scene classes in three macro-classes: Indoor, Nature, and Urban. The advance of the available datasets for scene recognition has significantly impacted scene recognition methods. Previously, methods focused on enhancing specific features, e.g. semantic features (López-Cifuentes et al., 2020), multi-layer features (Liu et al., 2019), or multiview features (Seong et al., 2020), achieving state-of-the-art results on MIT67 and SUN397 (Zeng et al., 2021). Methods trained on Places365, however, leverage the large amount of data so that standard CNNs significantly outperform previous approaches (Zhou et al., 2018).

Since SSPictR is primarily designed for robotics applications, such as visual navigation, our focus is particularly on indoor scene recognition. Pal et al. (2019) and Chen et al. (2019) proposed subsets of the Places365 dataset containing seven and 14 classes, respectively. Building on this,

Miao et al. (2021) proposed a novel model that resembles our approach by integrating knowledge from semantic segmentation maps: object-to-scene (OTS). OTS extracts object features through a pre-trained segmentation model and calculates object relations, outperforming both Pal et al. (2019) and Chen et al. (2019). However, OTS also requires up to 255 million model parameters and, therefore, only achieves an inference speed of three fps. More recently, attentional graph convolutional network (AGCN; Zhou et al. 2023) achieved higher performance than OTS on both datasets while also increasing inference time to 27 fps. Song & Ma (2023) proposed a semantic region relationship model (SRRM) and combined it with the PlacesCNN module by Zhao et al. (2023), yielding CSRRM, which achieves the current state-of-the-art performance on Places365-7 and Places365-14. In follow-up work (Song et al., 2024), the authors focused on computational efficiency that is essential for the low-resource and high-speed requirements of edge devices in practical robotics applications. However, their method is specific for enhancing scene recognition models and does not transfer to other tasks.

**Human Alignment.** Aligning representations of deep neural networks (DNNs) to humans is a promising avenue to increase performance and generalisabilty of computer vision models (Sucholutsky et al., 2023; Chang et al., 2019). While many works analysed object representations and their alignment to human similarity judgements (Muttenthaler et al., 2022; Geirhos et al., 2021; Hebart et al., 2020; King et al., 2019), meaningful representations of full scenes remain under-explored. Groen et al. (2017) analysed scene recognition in humans and found that in addition to object co-occurrence statistics as found by Stansbury et al. (2013), other features across different levels of visual processing play an important role, such as spatial layouts, boundaries, and textures. This encourages building scene representations based on object-level statistics, such as the presence and features of certain objects, but also highlights the importance of additional spatial information. Hence, we construct SSPictR from segmentation maps of images, which provide both object information as well as their spatial layout in the scene.

**Visual Complexity.** Segmentation maps and object-level statistics were also successfully used to predict the visual complexity of images (Nath et al., 2024). Visual complexity ratings of images are important for cognitive science studies, as they impact attention, engagement, and memorability of stimuli. Furthermore, visual complexity is relevant for practical applications, such as user experience on webpages and brand logo design (Kyle-Davidson et al., 2023). To understand and effectively model visual complexity, most methods rely on hand-crafted features to explain human complexity ratings, i.e., number of regions, frequency factor, and colours (Corchs et al., 2016), clutter and patch symmetry (Kyle-Davidson et al., 2023), or number of classes and segments (Nath et al., 2024). Supervised methods achieve the highest correlation with human ratings, but they require large datasets for training, where currently only IC9600 (Feng et al., 2023) is publicly available, offering 9,600 images across eight categories. Smaller datasets include Savoias (Saraee et al., 2020) consisting of 1,400 images with seven categories and VISC (Kyle-Davidson et al., 2023) containing 800 images across 12 sub-categories.

## 2.2 VECTOR SYMBOLIC ALGEBRAS

Vector symbolic algebras (VSAs) play an important role in cognitive architectures (Kleyko et al., 2023; Stewart et al., 2012) and showed improved performance of machine learning methods, e.g., ego-motion prediction (Mitrokhin et al., 2020), speech recognition (Imani et al., 2018), or object classification (Gallant & Culliton, 2016). VSAs offer a unique way of encoding symbolic meaning in hyper-dimensional distributed representations, making them inherently interpretable (Mitrokhin et al., 2020) and robust to errors (Rahimi et al., 2016). Furthermore, their potential application in neuromorphic hardware make them highly efficient, achieving speed gains of up to 100 times GPU performance (Blouw et al., 2019). The potential of VSAs in replicating cognitive maps for navigation has been shown via path integration (Dumont et al., 2022) and reinforcement learning on navigation tasks (Bartlett et al., 2022), while the potential in scene understanding has been shown via visual question answering (Komer et al., 2019; Penzkofer et al., 2024). For more application examples and a comprehensive literature review on VSAs, we refer to Kleyko et al. (2022; 2023).

In this work, we build upon the semantic pointer architecture (SPA) (Eliasmith, 2013), a cognitively-inspired VSA, which uses holographic reduced representations (HRRs) (Plate, 1995), i.e., a set of operations that can be applied for manipulation of hyper-dimensional vectors representing symbols. Semantic similarity of vectors is calculated by the dot product. Vectors can be bundled to represent

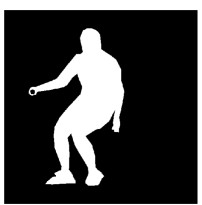 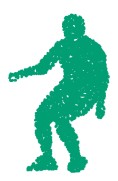 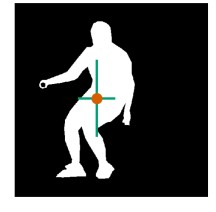 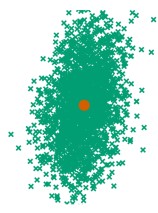 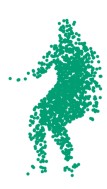

(a) Uniform sampling.  (b) Gaussian sampling.

Figure 2: Comparison of sampling methods on the same segmentation mask. For uniform sampling (a), a fixed percentage of points is drawn from the given mask. For Gaussian sampling (b), the centre of mass (orange) is computed. Then, a fixed number of samples from Gaussian distribution with mean as centre of mass and covariance of segmentation mask are drawn.

multiple concepts, computed by vector addition, and they can be bound together to represent concepts that belong together, e.g. a red apple. Binding – denoted by $\circledast$ – is circular convolution in HRR and the inverse operation (unbinding) is binding with a vector's pseudo-inverse. Komer et al. (2019) have introduced fractional binding, i.e. binding the vector with itself $k \in \mathbb{R}$ times, to encode continuous data, which enables the encoding of spatial locations.

## 3 METHOD

### 3.1 THEORETICAL BACKGROUND

Under the SPA, spatial semantic pointers (SSPs) were proposed for spatial representations through the following encoding scheme:

$$\phi(x) = \mathcal{F}^{-1}\{e^{i\lambda^{-1}Ax}\}, \tag{1}$$

where $\phi : x \in \mathbb{R}^2 \mapsto \mathbb{R}^d$, $\lambda$ defines the length scale of the encoded representation, $\mathcal{F}^{-1}$ denotes the inverse Fourier transform, and $A \in \mathbb{R}^{d \times 2}$ is a phase matrix whose columns consist of phasors representing different frequencies. For real-valued spatial representations, the phase matrix is conjugate symmetric. For biological realism of the SSP representation, further constraints are applied to the phase matrix that enable the replication of grid cell firing patterns as seen in Dumont & Eliasmith (2020). This is achieved by setting triplets of rows in the phase matrix $120°$ apart, resulting in gridded interference patterns. As given by biological experimental findings, this data lies on the hypertoroidal manifold (Gardner et al., 2021). The dimensionality $d$ is given by $d = n_{\text{scales}} \cdot n_{\text{rotates}} \cdot 3 \cdot 2 + 1$, where $n_{\text{scales}}$ denotes the scale of the firing pattern activity, $n_{\text{rotates}}$ denotes the orientation of the grid cells, 3 denotes triplets, 2 for conjugate symmetry, and $+1$ for the 0-frequency term.

Using the SSP representation and the set of operations given by HRR, we can construct cognitive maps, for example, we can construct the map M of a cat at location $(x_1, y_1)$, a mouse at location $(x_2, y_2)$, and cheese at location $(x_3, y_3)$:

$$\mathbf{M} = \mathbf{CAT} \circledast \phi(x_1, y_1) + \mathbf{MOUSE} \circledast \phi(x_2, y_2) + \mathbf{CHEESE} \circledast \phi(x_3, y_3), \tag{2}$$

then, to query an object's location, unbinding can be used :

$$\mathbf{M} \circledast \mathbf{CHEESE}^{-1} = \phi(x_3, y_3) + noise \tag{3}$$

This noise is due to the nature of the unbinding operation: since the unbinding of **CHEESE** distributes over the map representation, $\mathbf{CAT} \circledast \phi(x_1, y_1) \circledast \mathbf{CHEESE}^{-1}$ and $\mathbf{MOUSE} \circledast \phi(x_2, y_2) \circledast \mathbf{CHEESE}^{-1}$ produces random noise.

### 3.2 ENCODING SCHEME

In previous works, objects have been encoded as point sources (Dumont et al., 2022), or a set of bounding box coordinates (Penzkofer et al., 2024). However, for a more accurate representation, we instead encode segmentation maps of a scene as follows:

$$\mathbf{S} = \sum_i \left[ \text{obj}_i \circledast \int_{A_i} \phi(x)dx \right] \tag{4}$$

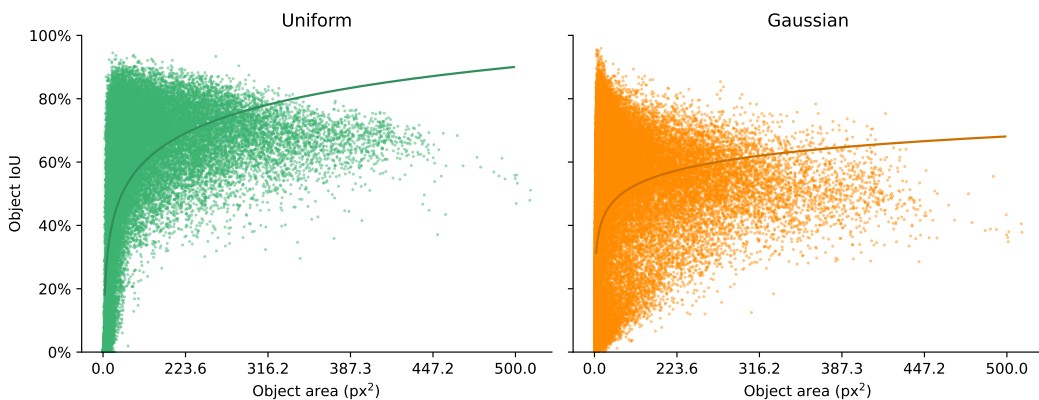

Figure 3: Object-level IoU dependent on object area – comparison between encoding schemes: uniform encoding (left) and Gaussian encoding (right).

In this hyper-dimensional representation, $\text{obj}_i \in \mathbb{R}^d$ is a semantic pointer (SP), representing the class of a given object $i$. This object is bound with a bundle of SSPs representing the area occupied by this object in the representation. In our preliminary analysis (see Table 1) we found that only a percentage of the pixels of a given object can be encoded without a significant loss in representation accuracy. To this end, we sample the representation either via uniform sampling within the mask or by sampling from a Gaussian distribution with the mean set as the mask's centre of mass and covariance matrix given by the segmentation mask. Points outside the mask are not encoded. Both sampling methods are visualised in Figure 2, where we see that the Gaussian sampling technique focuses on the center of the object.

To evaluate the quality of the SSP representation, we decode the masks used to generate the scene representation $\mathbf{S}$. We calculate the similarity map of an object $j$ in the scene by taking the dot product between a grid of SSPs and the encoded scene bound with the inverse of that object's SP: $\left\langle S \circledast \text{obj}_j^{-1}, \int_{A_{\text{grid}}} \phi(x)dx \right\rangle$. This yields an approximate similarity value between the SSP grid $A_{\text{grid}}$ that represents each possible location and the scene SSP $\mathbf{S}$ that is queried for the object of interest. The set of similarity values larger than some threshold $\tau$ is used as the decoded mask. Finally, the intersection-over-union (IoU) is calculated using the ground truth and the decoded mask. This approach can be used to probe for every object in the scene, to determine what is at a specific location, or to verify that an object exists in the scene, making this representation inherently interpretable. We then optimise the hyperparameters most important for this encoding, i.e., the length scale, threshold, percentage of encoded points, and SSP dimensionality.

Table 1 shows the average number of encoded points, the average encoding time, and the average object IoU for each best representation configuration with tuned lengthscale $\lambda$ and threshold $\tau$ parameters. For details on finetuning the parameters to find the best configuration, see Figure 5 in the Appendix. We perform a grid search on 50 samples from the COCOStuff (Caesar et al., 2018) dataset, which encodes $8.34$ objects on average. Increasing the SSP dimensionality also increases the average decoding accuracy for all encoding schemes. Similarly, encoding time also increases with SSP dimensionality, but the number of encoded points does not. This is expected as the point selection only depends on the encoding scheme and the size of the object masks. The increase in encoding time is due to the larger dimensionality of all vectors $x$ in equation 4. Most interestingly, we find that uniform sampling is on par with the full encoding scheme while being significantly faster. Gaussian sampling achieves the best results in terms of IoU and encoding time as it uses a fixed number of samples to draw, which increases decoding accuracy for smaller objects.

Based on this preliminary analysis, we select $3,751$ dimensions for the SSPictR representation. We strive to keep the representation as compact as possible, i.e., at the lowest SSP dimension that achieves reasonable results. This allows for a compression rate of only $0.46$ bits per pixel for a 512x512 image, similar to the highest possible compression rate of JPEG (Dotzel et al., 2024). However, as both sampled encoding schemes achieve similar results in terms of IoU, we further evaluate them. To this end, we analyse the effect of object areas on the IoU accuracy, where a weak

Table 1: Hyper-parameter analysis on 50 samples of the COCO-Stuff dataset. Uncertainty terms correspond to one standard deviation.

| Encoding | Dimensions | $\lambda$ | $\tau$ | # Points $\downarrow$ | Time [s] $\downarrow$ | IoU $\uparrow$ |
|---|---|---|---|---|---|---|
| Full scene | 1,015 | 27.5 | 0.65 | 179,950 | 8.8±1.1 | 36.1±12.9% |
|  | 1,945 | 30.0 | 0.7 | 179,950 | 37.7±4.8 | 43.4±13.1% |
|  | 3,751 | 22 | 0.7 | 179,950 | 40.1±5.1 | 47.6±13.5% |
| Uniform | 1,015 | 27.5 | 0.65 | 3,706 | 1.5±0.3 | 37.5±13.1% |
|  | 1,945 | 30 | 0.7 | 3,706 | 6.0±1.1 | 41.8±12.5% |
|  | 3,751 | 22 | 0.7 | 3,706 | 6.3±1.1 | 47.1±13.6% |
| Gaussian | 1,015 | 20.0 | 0.6 | 16,202 | **2.2±0.4** | 40.5±9.4 % |
|  | 1,945 | 15.0 | 0.6 | 16,202 | 4.7±1.4 | 45.3±9.3 % |
|  | 3,751 | 17 | 0.55 | 16,202 | 5.1±1.5 | **47.9±9.5 %** |

correlation was reported in (Lu et al., 2019; Penzkofer et al., 2024). Figure 3 shows that the uniform encoding scheme struggles with small objects, i.e., $< 200px^2$, but achieves higher object IoU in general compared to the Gaussian encoding. The fitted logarithmic curves with $-0.72+0.13*log(x)$ (Uniform) and $-0.15 + 0.07 * log(x)$ (Gaussian) emphasise this point. Hence, we selected uniform encoding for all analyses that follow.

## 4 EXPERIMENTS

We first evaluate the representation quality by calculating the reconstruction accuracy for semantic segmentation on the COCOStuff 2017 dataset (Caesar et al., 2018) with 118K images. Furthermore, we perform linear probing by training a small scene recognition model on a reduced Places365 dataset (Zhou et al., 2018) and test generalisation for out-of-domain (OOD) scene recognition on ADE20K (Zhou et al., 2017). As an additional downstream application, we predict the visual complexity of images with SSPictR as features for a regression model.

### 4.1 REPRESENTATION QUALITY

For evaluating the reconstruction accuracy, we compute all SSPs with 3751 dimensions and the uniform encoding scheme for the COCOStuff train and validation set (Caesar et al., 2018). Then, we reconstruct the segmentation maps by unbinding the scene representation with each object vector separately, yielding the similarity map for the queried object. We select all points above $\tau = 0.7$ similarity to belong to the queried object. We compare this mask with the ground truth mask of the object and calculate the pixel-wise IoU. With the uniform encoding scheme, we achieve $45.36\% \pm 13.05\%$ average IoU on the COCOStuff validation split. Here, we encode 3,770 points on average with an encoding time of $6.13 \pm 1.18$s per image. We further increase the reconstruction accuracy by training a UNet model (Ronneberger et al., 2015) to refine the object masks based on the SSP similarity maps instead of using the threshold $\tau$.

Our UNet model consists of four encoder and decoder layers with a total of 465K trainable parameters. We trained the model on a subset of the COCOStuff 2017 train dataset Caesar et al. (2018), i.e. we use 10K samples of similarity maps, which amounts to a total of 1,235 training images, split into train and validation. We evaluate the model on the official validation split (not used for validation during training) of 5,000 images and achieve an average IoU of $57.3 \pm 7.1\%$, a considerable 13.2% increase. As shown in Figure 4, the model learns to refine the given similarity maps, removing excess noise. Based on further examples, we find that the model learns to adapt the threshold of the similarity map based on object-specific features, mainly size, as the SSPictR representation struggles with small objects (see Figure 3). To conclude, while we only trained on a small subset of the available dataset, we achieved a significant reconstruction performance increase. We believe this could be further increased by more training or a more elaborate network structure.

We then evaluate the quality of our representation via linear probing, a standard method for determining intermediate representation quality in self-supervised models (Mu et al., 2022). More specif-

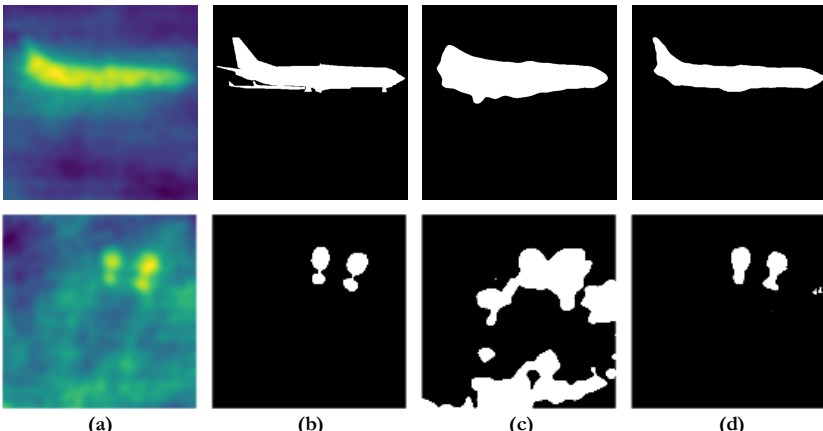

|  (a)  |  (b)  |  (c)  |  (d)  |

Figure 4: Example **(a)** similarity maps with **(b)** ground truth masks , **(c)** threshold prediction, and **(d)** refined Unet model prediction.

ically, we train a small classification model on the SSP representations of the popular Places365-7 (Pal et al., 2019) and Places365-14 (Chen et al., 2019) subsets of the Places365 dataset Zhou et al. (2018) for indoor scene recognition. We follow the setup by Miao et al. (2021). The Places365-7 training set consists of 35,000 images, and the Places365-14 training set of 55,000 images. As Places365 does not offer segmentation maps, we run the pre-trained VPD model (Zhao et al., 2023) for semantic segmentation, which was trained on the ADE20K dataset. Therefore, we use the same object classes for both datasets. After generating the 3,751-dimensional SSPs for all samples, we train a two-layer linear neural network (NN) model with approx. seven million parameters. The linear NN takes the SSPs as input and has a hidden dimension of 1,875, ultimately reducing the features to the output dimension of seven or 14 classes. We use both batch normalisation and dropout layers, where the batch size is set to 1,024 and $p_{\text{dropout}} = 0.4$. We use AdamW (Loshchilov & Hutter, 2018) as an optimiser with a learning rate of $0.00195$. We train the model for 25 epochs and evaluate its performance on the held-out official validation set with 700 images and 1,100 images for the seven and 14 classes, respectively.

Table 2 summarises our scene recognition results, also in comparison with previous methods. As can be seen from the table, our method achieves comparable performance in terms of accuracy but only takes the 3,751-dim SSP vectors as inputs and achieves faster inference speed of 2,875 frames-per-second (fps), 25 times more than the next best method. This showcases the potential of SSPictR to be used as a low-memory, high-efficiency image representation for edge devices. We further evaluate the generalisation performance of our trained Places365-7 model on the ADE20K dataset as OOD data and achieve a remarkable performance of 94.5% classification accuracy. This even outperforms our SVM baseline (RBF kernel with $c = 5$) trained on ADE20K directly, achieving 94.2% test accuracy on a hold-out set of 817 images. The higher performance on the ADE20K dataset might be explained by the availability of ground truth segmentation maps. The quality of segmentation maps is a limiting factor of current scene recognition methods (Song et al., 2024).

Table 2: Scene recognition results and comparison to previous state-of-the-art.

| Method | Parameters | Inference [fps] | Places365-7 | Places365-14 |
|---|---|---|---|---|
| OTS Miao et al. (2021) | 255 M | 3 | 90.1 | 85.9 |
| AGCN Zhou et al. (2023) | 85 M | 27 | 91.7 | 86.0 |
| CSRRM Song & Ma (2023) | 50 M | - | **93.4** | **88.7** |
| GLS + BCL Song et al. (2024) | 25 M | 115 | 90.6 | 86.6 |
| SSPictR (Ours) | **7 M** | **2,857** | 90.1 | 82.2 |

Table 3: Visual complexity prediction and comparison to other handcrafted methods, as well as supervised methods trained on larger datasets. We report Spearman rank correlation $r$ with human complexity ratings.

| Method | Savoias Art | Savoias Scenes | Savoias Int. Design | VISC |
|---|---|---|---|---|
| **handcrafted features** | | | | |
| clutter + symmetry Kyle-Davidson et al. (2023) | 0.55 | 0.54 | 0.74 | 0.60 |
| #seg + #classes Nath et al. (2024) | 0.73 | 0.78 | 0.61 | 0.56 |
| #seg + #classes + symmetry Nath et al. (2024) | - | - | 0.80 | 0.68 |
| **supervised models** | | | | |
| ComplexityNet Kyle-Davidson et al. (2023) | 0.30 | 0.36 | 0.56 | - |
| ICNet Feng et al. (2023) | **0.81** | **0.79** | **0.89** | **0.72** |
| SSPs | 0.42 (KR) | 0.52 (RF) | 0.45 (KR) | 0.53 (RF) |
| SSPs + # classes | 0.45 (KR) | 0.60 (KR) | 0.51 (RF) | 0.54 (RF) |
| SSPs + symmetry | 0.45 (RF) | 0.54 (RF) | 0.49 (RF) | 0.57 (RF) |

For a further visual analysis of the SSP representations from the ADE20K dataset in comparison to Places365, see Appendix A.2.

## 4.2 VISUAL COMPLEXITY

To evaluate the generalisability of SSPictR to other downstream tasks, we predict human visual complexity ratings of images. Following previous work (Nath et al., 2024) we use handcrafted features, i.e., our SSP representations, and train simple regressors to predict the complexity scores that go from zero to one hundred. We evaluate the following models: support vector regression (SVR), kernel ridge regression (KR), gradient boosting (GB), and random forest regression (RF). KR with a cosine kernel, i.e., cosine similarity as a comparative measure between different vectors, yields the best results. This is intuitive as the cosine similarity has been shown to be the same as the distance between two unitary SSPs in the Fourier domain (Voelker, 2020). We evaluate our models on 3 classes of the Savoias dataset (Saraee et al., 2020) and on the full VISC dataset with a 7-fold cross-validation. Unfortunately, we did not get access to the larger scale IC9600 (Feng et al., 2023) dataset and are therefore unable to test a supervised method. We perform the same preprocessing pipeline as for the scene recognition model on Places365; first, we generate segmentation maps for each image with the pre-trained VPD model (Zhao et al., 2023), then, we compute our SSP representations with the uniform encoding scheme. For a visualisation of samples from the different datasets, see Appendix A.3. The art category (Savoias) is the most difficult for the VPD model to segment, as it is comprised of paintings or simple drawings, which was not in the training data (ADE20K; Zhou et al. 2017) of VPD. Further, we found that the interior design category, while close to the interior scene images in ADE20K, has significantly more objects encoded per image: 22.26 on average compared to 10.87 (VISC, Savoias SCENES) and 8.23 (Savoias ART). Additionally, the average object size is also significantly smaller.

Table 3 summarises relevant prior work with Spearman rank correlation coefficient $r$ between predictions and the human complexity ratings (values taken over from Nath et al. (2024)). Our results show that we achieve reasonable performance on the Savoias SCENES dataset, i.e., on par with Kyle-Davidson et al. (2023) and similarly on VISC. However, SSPs as features significantly struggle with the ART and INTERIOR DESIGN category; we believe this is due to limited segmentation accuracy (ART) and the high number of objects (INTERIOR DESIGN). Further, we test whether additional features can increase the correlation coefficient. Inspired by Nath et al. (2024), we use the number of classes extracted from the number of unique labels in the segmentation map. Additionally, we also calculate the patch symmetry, as proposed by Kyle-Davidson et al. (2023). Both features improve the predictions, however, since the added feature is only one of 3752 the impact is not significant. Overall, we show that SSPictR as features for visual complexity prediction perform

similar to previous handcrafted features, but there is still room to improve. We believe a supervised method is better suited to work with the large feature space of 3751 dimensions, and we will test this hypothesis when a suitable dataset becomes available.

## 5  DISCUSSION AND CONCLUSION

We have presented SSPictR – a cognitively-inspired image representation that is inherently interpretable and efficient. We evaluated different encoding schemes and found the best hyperparameters for encoding a full scene based on segmentation maps. Further, we evaluated the representation quality by calculating the reconstruction accuracy on COCOStuff (Caesar et al., 2018), where we found that a simple mask refinement model can significantly enhance reconstruction IoU to $57.3 \pm 7.1\%$. The key advantage of SSPictR is that the representations are compact, only requiring 0.46 bits per pixel and that they can directly be used for downstream tasks, such as scene recognition and visual complexity prediction.

For scene recognition, a key task for robotic navigation (Xie et al., 2020; Miao et al., 2021), we trained a small neural network that achieved comparable performance on two indoor scene subsets of the popular Places365 dataset (Zhou et al., 2018). Similar to previous work (Song et al., 2024), we found that scene recognition accuracy is limited by segmentation quality, i.e. where ground truth segmentation maps are available, we achieved a remarkable scene recognition accuracy of 94.2% with a SVM on the ADE20K dataset (Zhou et al., 2017). Further, the model trained on Places365 achieves an even higher performance on ADE20K as out-of-distribution data with 94.5%, highlighting the generalisation ability of our representation. Additionally, SSPictR achieves 25 times higher processing speed than previous methods.

Moreover, SSPictR is able to generalise across different tasks, as we have shown by performing visual complexity prediction. Visual complexity prediction is an important task in cognitive science, as perceived complexity is linked to engagement and attention, influencing subjects' reaction to given stimuli (Nath et al., 2024). While we achieve comparable performance to other handcrafted feature methods on datasets that represent scenes, i.e., VISC (Kyle-Davidson et al., 2023) and Savoias SCENES (Saraee et al., 2020), the SSP representations struggle with abstract images (Savoias ART) and images with a significantly higher amount of objects (Savoias INTERIOR DESIGN). The latter could potentially be addressed by increasing the capacity of the SSPs by allowing a higher dimensionality. However, the more abstract art images do not encompass real-world scenes with spatial layouts, which is the intended application area of SSPictR.

Our overall goal is to develop a compact image representation that can be deployed end-to-end on edge devices for efficient visual navigation and other downstream tasks. SSPictR is an important first step towards this goal. In future work, we would like to address some of the discussed limitations by integrating a larger object vocabulary and more fine-grained segmentation, e.g., by using the SAM model (Kirillov et al., 2023), which detects segments at different scales. Further, we plan to train an end-to-end image to SSP model, potentially with a fully neural implementation to achieve the highest efficiency. In general, we believe SSPictR is uniquely suitable for robotics applications, such as embodied visual navigation (Yadav et al., 2023; Ramakrishnan et al., 2021), as well as for further cognitive applications. More specifically, we would like to analyse the alignment of SSPictR to human scene representations, e.g., by performing similarity analysis on fMRI data (Chang et al., 2019), or integrate it into symbolic reasoning systems (Hersche et al., 2023).

As we have shown, SSPictR opens up a new direction of research, leveraging the advantages of cognitively-inspired image representations.

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

# A APPENDIX

## A.1 LENGTHSCALE FINETUNING

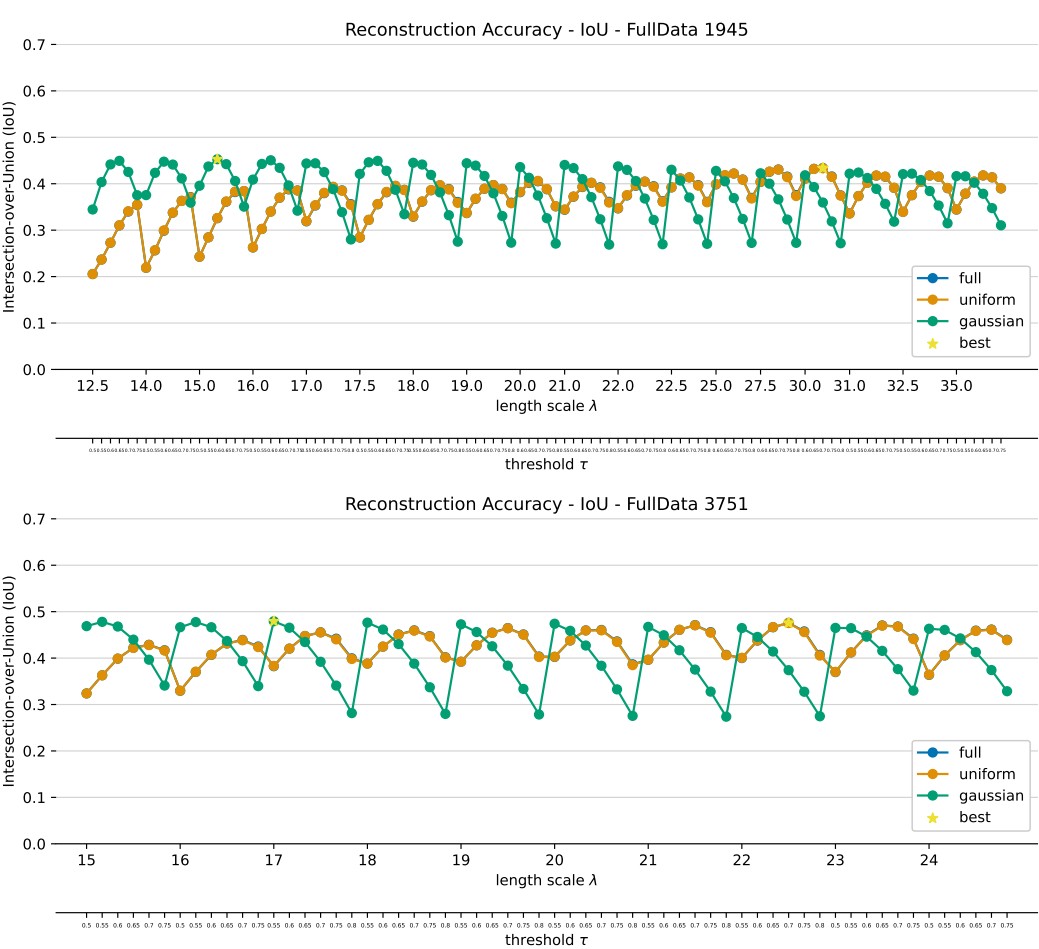

Figure 5: Lengthscale $\lambda$ and threshold $\tau$ fine-tuning on 50 samples of COCOStuff (Caesar et al., 2018) dataset.

For fine-tuning the lengthscale $\lambda$ parameter, we performed a grid search on 50 samples from the COCOStuff dataset (Caesar et al., 2018). Results are presented in Figure 5. We encoded all objects with the full, uniform, or Gaussian encoding scheme and evaluated reconstruction performance in terms of IoU between ground truth object segmentation and our predicted mask, i.e. taking all points above the threshold $\tau$ in the similarity map after unbinding with the object's inverse SSP. We also optimised $\tau$ (bottom of y-axis) and marked the best configuration with a star. Overall, the IoU accuracy increases with an increase in SSP dimensions, from 1945 (top) to 3751 dimensions (bottom). This is due to the increased capacity of higher dimensional vectors.

## A.2 SCENE RECOGNITION - DATA COMPARISON

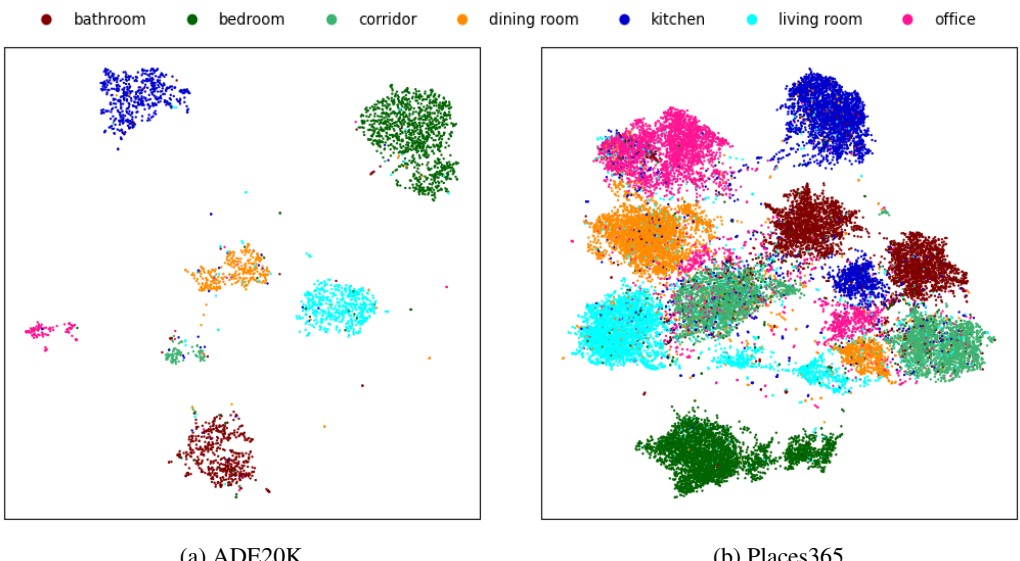

(a) ADE20K                                      (b) Places365

Figure 6: UMap (McInnes et al., 2020) visualisation of SSP representations for 7 indoor classes in (a) ADE20K (Zhou et al., 2017) and (b) Places365 dataset (Zhou et al., 2018).

Here we compare our SSP representations regarding discrimination quality for the 7 indoor scene classes: bathroom, bedroom, corridor, kitchen, living room, office, and dining room. Both on the Places365-7 dataset (Zhou et al., 2018) and the ADE20K dataset (Zhou et al., 2017). We performed dimensionality reduction via the UMAP algorithm (McInnes et al., 2020) to visualise the representations on both datasets (see Figure 6). We can directly see that ADE20K offers significantly fewer samples with only 4497 images compared to 35000 in the Places365 training set. Despite the availability of more data for Places, it was more difficult to separate the SSP representations according to the correct classes. This might be due to wrong segmentations on the Places dataset. Specifically, some objects might be mislabeled if they do not exist in the ADE20K data. This separation difficulty explains the lower scene recognition performance of our method on Places compared to ADE20K, where a simple SVM classifier is able to get 94% accuracy. Further, this implies that some additional features or better segmentations might be necessary to improve classification performance on Places365.

## A.3 VISUAL COMPLEXITY - DATA COMPARISON

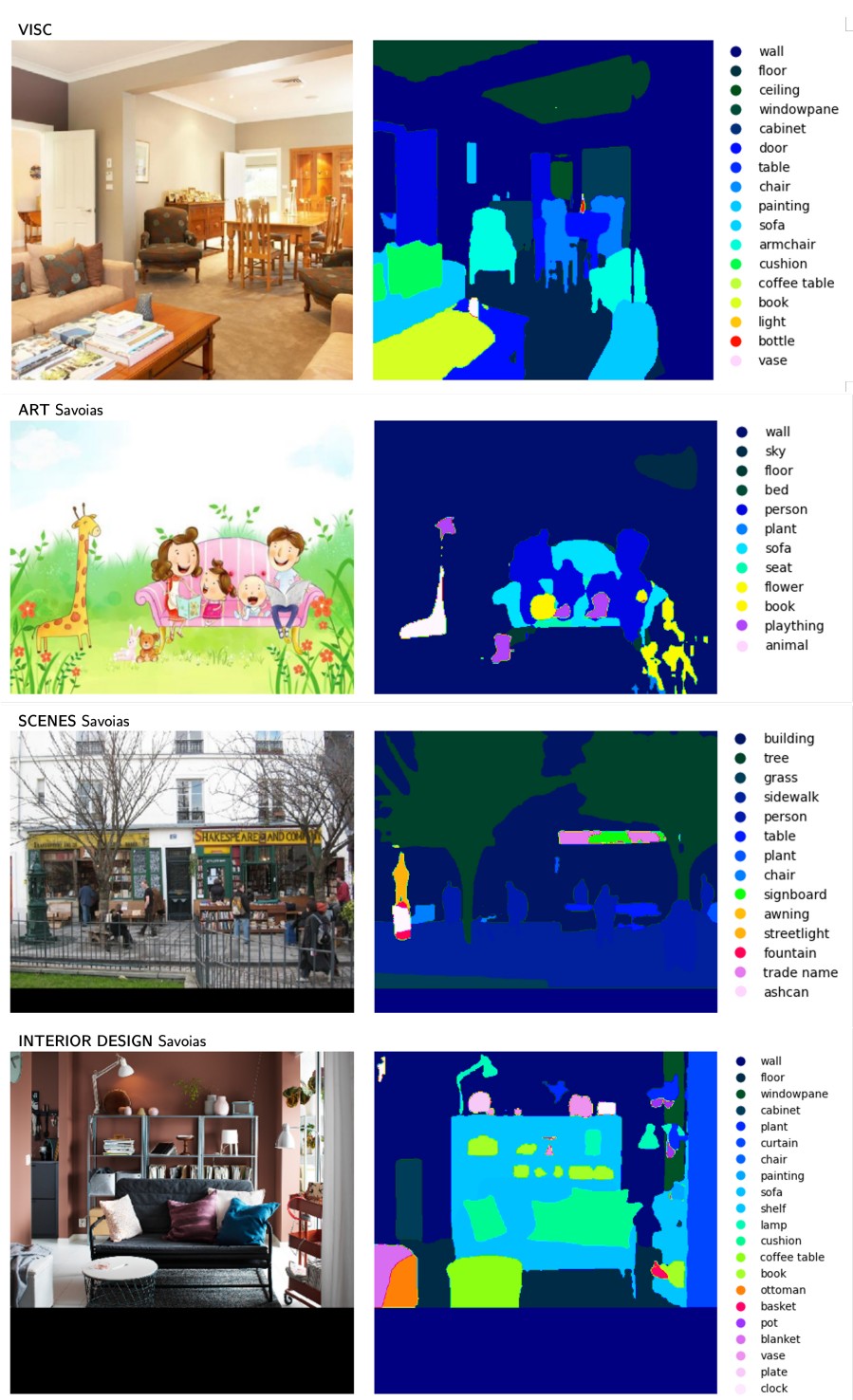

Figure 7: Comparison of images and generated segmentation maps between VISC (Kyle-Davidson et al., 2023) and Savoias (Saraee et al., 2020) three classes: art, scenes, and interior design.

