# OpenReview forum: "SSPictR: Spatial Semantic Pointer Picture Representation"
_ICLR.cc/2025/Conference — ICLR 2025 Conference Withdrawn Submission_

### Official Review · Reviewer_NvbL · 2024-11-03

**Soundness:** 1
**Presentation:** 3
**Contribution:** 2
**Rating:** 3
**Confidence:** 3

**Summary:**

The approach consists of 3 models:
1. VPD (255M parameters) - trained on the ADE20k dataset, takes an image as input and produces semantic segmentation masks.

2. U-Net (365K parameters) trained on the COCOstuff dataset, takes an image and semantic segmentation masks from VPD as input, and produces refined segmentation masks

3. Untrainable module: SSPictR-encoding-scheme converts segmentation masks from U-Net to compressed scene vector representation SSPictR-vector of size 3751

3. SSPictR model (7M parameters, 2857 fps) trained on Places-365 (for 7 and 14 classes), takes as input the scene vector representation from SSPictR-encoding-scheme, and produces a scene class according to the Places365 datasets for 7 and 14 classes as output

The final result of the approach is tested on datasets:
- Places-365 and ADE20k for scene recognition task
- COCOStuff for segmentation reconstruction task from SSPictR-vector of size 3751 (compressed scene vector representation)
- VISC and Savoias for prediction of visual complexity task


The proposed approach has accuracy comparable to the OTS and GLS approaches, but is less accurate than the AGCN and CSRRM approaches. The proposed approach is orders of magnitude faster than others (but the speed is measured only for the last SSPictR-model, excluding VPD and U-Net models), Table 2.

**Strengths:**

1. The SSPictR model is trained on the Places-365 dataset and tested on the ADE20k dataset, which is out-of-domain and shows the generalizability of the approach.

2. The speed of the SSPictR model itself (7M parameters) is quite high 2857 fps, Table 2.

**Weaknesses:**

1. Approaches that receive different representations as input are compared. Only the speed of SSPictR is measured (the last part of the whole approach is measured), without taking into account VPD and U-Net. While for a fair comparison, identical input data should be used for all approaches, or the execution times of the approaches as a whole, including the intermediate networks (VPD and U-Net) should be compared.

It is possible that the high accuracy of SSPictR is partly due to the use of more accurate but slower VPD and U-Net models to generate the segmentation masks that are the input to SSPictR.
Approaches such as OTS, AGCN, and CSRRM use simplified intermediate networks or features that may be less advanced in segmentation but are faster. This reduces their accuracy compared to SSPictR.
Thus, the high accuracy of SSPictR may indeed be partly due to the use of more accurate but slower intermediate models (VPD and U-Net) for segmentation, which is different from other approaches that make the adjustment for speed.


2. The proposed approach is not more accurate than other approaches, Table 2.

**Questions:**

1. Could it be that the high accuracy of the SSPictR model is achieved by using more accurate models such as VPD and U-Net, which are significantly slower than similar intermediate networks in the compared OTS, AGCN, and CSRRM approaches?

2. Could the whole approach you proposed (SSPictR + VPD + U-Net) be slower than comparable approaches as a whole (which also include intermediate networks)?

---

### Official Review · Reviewer_uFRi · 2024-11-04

**Soundness:** 4
**Presentation:** 3
**Contribution:** 2
**Rating:** 3
**Confidence:** 3

**Summary:**

This paper introduces SSPictR, a cognition-inspired image representation based on spatial semantic pointers (SSPs). SSPictR encodes semantic labels and their spatial locations extracted from segmentation maps, obtaining a single vector to capture a fully decodable neuro-symbolic representation of a natural scene. The authors claim that the representation is inherently interpretable, offers a high compression rate, and provides significantly faster inference speeds for downstream tasks, such as scene recognition. The experimental results demonstrate the generalizability of SSPictR across multiple tasks and datasets, including Scene recognition on the Places365 and ADE20K datasets, Segmentation reconstruction on the COCOStuff dataset, Prediction of visual complexity on the VISC and Savoias datasets.

**Strengths:**

1. The representation uses a single vector to capture the entire scene, resulting in a high compression rate while the semantic segmentation reconstruction is efficient and the results seem acceptable according to the experiments.

2. SSPictR is designed to be inherently interpretable, allowing for a clear understanding of the encoded information. It is different from the current pure learning-based representation methods which require a large amount of the training samples.

3. SSPictR is a neuro-symbolic representation scheme that has good potential for various high-level tasks that require symbolic computation.

4. The paper is well written and it is easy to follow the main idea of the paper.

**Weaknesses:**

1. Although SSPictR performs well in applications, its core technology is still based on the existing SSP framework and the technical innovations seem limited. Thus, it is quite difficult for me to agree that the paper meets the standards of technical innovation required by ICLR.

2. SSPictR provides a highly compact but meaningful representation, which achieves significantly faster inference speeds compared to existing methods since it only needs a small model (MLP) in the test phase. However,  I do not think the 7M model parameters reported in Table 2 to be reasonable. The representation of each image heavily relies on the segmentation model, which must be used online during the test phase. Therefore, the total parameter count for the scene recognition model should include both the segmentation model and the recognition model. Consequently, the reported inference speed in the experiments is not convincing, as it does not account for the computational overhead of the segmentation model. It would be beneficial to provide more detailed information about the model sizes, specifically separating the sizes of the segmentation model and the recognition model. Additionally, it would be useful to report FPS which can be achieved when performing both segmentation and recognition. This information would give a clearer picture of the practical feasibility and efficiency of the proposed method.

3. The performance of SSPictR is highly dependent on the quality of the segmentation maps. In other words, its effectiveness relies on the segmentation model used during the inference stage, specifically in terms of the number of classes and the model performance. Therefore, although SSPictR is referred to as a cognition-inspired model, its flexibility is still constrained by upstream traditional black-box convolutional models, limiting its adaptability for downstream tasks. It is better to discuss how the choice of segmentation model impacts SSPictR's performance and flexibility.

**Questions:**

For a neuro-symbolic representation, I think SSPictR should have a more fundamental baseline for scene recognition. Specifically, each semantic category should be represented by a one-hot vector, forming a codebook of all semantic categories. Each image then could be encoded directly using this codebook, with a 1 in the corresponding dimension if the category is present and a 0 otherwise. Such a scheme is reasonable because the proposed method relies on a semantic map, which already embeds semantic information. Essentially, the scene in a specific image is a combination of the objects within it. The baseline described above incorporates semantic information but ignores spatial information. Therefore, it can serve as a baseline representation to validate the effectiveness of the spatial information embedding in SSPictR.

**Details Of Ethics Concerns:**

This paper introduces SSPictR, a cognition-inspired image representation method. All of the experiments focus on standard computer vision tasks, and therefore, I believe no ethical review is required.

---

### Official Review · Reviewer_p2Dz · 2024-11-07

**Soundness:** 4
**Presentation:** 2
**Contribution:** 3
**Rating:** 6
**Confidence:** 2

**Summary:**

The authors present SSPictR which represents image data by encoding semantic labels and their spatial locations within a single high-dimensional vector. It operates via the semantic pointer architecture (SPA), which allows it to bind spatial coordinates with object labels into a single compact representation. This vector-based encoding makes it possible to reconstruct scenes and predict attributes like scene type and visual complexity with fewer parameters and high speed.

**Strengths:**

SSPictR reduces data size to 0.46 bits per pixel and achieves fast inference speeds of up to 2,857 frames per second, making it efficient and suitable for real-time tasks. Its generalisability allows it to perform well across tasks like scene recognition and visual complexity prediction, while its interpretable vector representation enables easy querying of objects within scenes.

**Weaknesses:**

1. SSPictR struggles to accurately represent small objects within scenes, which reduces its accuracy in tasks where small objects are essential.

2. Performance is highly dependent on the quality of segmentation maps. The representation suffers if segmentation is inaccurate.

**Questions:**

1. SSPictR has limitations in accurately representing small objects. Could the authors provide insights on potential adaptations or improvements to enhance its performance in such a scenario?

2. Has the model been tested on edge devices or lower-power hardware configurations commonly used in real-time applications, such as robotics or mobile devices? Understanding its performance on such hardware would clarify its practical applicability.

---

### Note · Authors · 2024-12-03

**Comment:**

We thank the reviewers for their detailed reviews. We believe we can address their concerns with an extension of the manuscript, which we are currently working on, but does not fit the scope of the rebuttal. Therefore, we withdraw the paper from the submission at ICLR.

**Withdrawal Confirmation:**

I have read and agree with the venue's withdrawal policy on behalf of myself and my co-authors.